# Extrusion-Based 3D Printing of Stretchable Electronic Coating for Condition Monitoring of Suction Cups

**DOI:** 10.3390/mi13101606

**Published:** 2022-09-27

**Authors:** Van-Cuong Nguyen, Minh-Quyen Le, Jean-François Mogniotte, Jean-Fabien Capsal, Pierre-Jean Cottinet

**Affiliations:** 1Laboratoire de Génie Electrique et Ferroélectricité, Institut National des Sciences Appliquées, Université de Lyon, 69621 Villeurbanne, France; 2Hybria Institute of Business and Technologies, Écully Campus, 69130 Écully, France

**Keywords:** design optimization, conductive coating, 3D printing additive manufacturing, printed electronics, condition monitoring, piezoresistive sensor, ink characterization

## Abstract

Suction cups (SCs) are used extensively by the industrial sector, particularly for a wide variety of automated material-handling applications. To enhance productivity and reduce maintenance costs, an online supervision system is essential to check the status of SCs. This paper thus proposes an innovative method for condition monitoring of SCs coated with printed electronics whose electrical resistance is supposed to be correlated with the mechanical strain. A simulation model is first examined to observe the deformation of SCs under vacuum compression, which is needed for the development of sensor coating thanks to the 3D printing process. The proposed design involves three circle-shaped sensors, two for the top and bottom bellows (whose mechanical strains are revealed to be the most significant), and one for the lip (small strain, but important stress that might provoke wear and tear in the long term). For the sake of simplicity, practical measurement is performed on 2D samples coated with two different conductive inks subjected to unidirectional tensile loading. Graphical representations together with analytical models of both linear and nonlinear piezoresistive responses allows for the characterization of the inks’ behavior under several conditions of displacement and speed inputs. After a comparison of the two inks, the most appropriate is selected as a consequence of its excellent adhesion and stretchability, which are essential criteria to meet the target field. Room temperature extrusion-based 3D printing is then investigated using a motorized 3D Hyrel printer with a syringe-extrusion modular system. Design optimization is finally carried out to enhance the surface detection of sensitive elements while minimizing the effect of electrodes. Although several issues still need to be further considered to match specifications imposed by our industrial partner, the achievement of this work is meaningful and could pave the way for a new generation of SCs integrated with smart sensing devices. The 3D printing of conductive ink directly on the cup’s curving surface is a true challenge, which has been demonstrated, for the first time, to be technically feasible throughout the additive manufacturing (AM) process.

## 1. Introduction

Today, manufacturers are looking to develop their products for added value and thinking about integrating functions to make the products “intelligent.” The idea of having an intelligent predictive organ was born of the need for predictive maintenance and the evolution of printed electronics [1,2]. In the industrial context, it is important to have the highest possible productivity to remain competitive, which implies that the machines used must be in operation as long as possible. However, their maintenance is crucial: an undetected anomaly can, to some extent, lead to unexpected shutdown of the tools and even production stoppage, with sometimes considerable damage to those not affected by the initial failure [3,4,5].

Among strategic tools, suction cups (SCs) are one of the most widely used in industrial sectors, such as packaging, plastics, food, sheet metal, robots, machine, or any other kind of automated process [6,7,8]. They are mostly exploited to handle or move fragile items, sometimes with huge amounts of weight. In its mode of action, the SC is subjected to vacuum that sucks out all of the air from the inside, and then is released after interruption of the vacuum. These actions are usually fast and repetitive in several cycles, resulting in drastically mechanical strain and stress to the SCs. Therefore, online condition monitoring of SCs is essential for a reduction of maintenance’s cost, allowing for failure detection and spare parts ordering in timely fashion. Predictive maintenance can be achieved in several ways, either by integrating sensors directly into the structure or coating them on its surface [9,10,11,12,13]. The first solution leads to more accurate measurement, as the sensor is usually mounted on the position where it needs to be sensed. However, this method is not applicable to all systems, because of its complicated integration and cumbersome nature. The second solution is electronic printed coating whose resistance variation is supposed to correlate with the mechanical strain of the structure, making early failure detection of SC possible. This approach is defined as direct printing of electronic components on monitored substrates through the combination of 3D printing processes and conductive/dielectric inks with more functionalities [14,15,16]. For instance, Thetpraphi et al. recently developed a full 3D-printed-force actuator based on an advanced electroactive polymer used for large and live optical mirror applications [17]. Grinberg et al. presented a lightweight, low-cost, and 4D-printing compatible piezoelectric composite that had high potential in smart-implant prostheses [18]. DeGraff et al. reported on a printable and low-profile strain sensor using multiwall carbon nanotube thin films called buckypaper [19]. Liu et al. proposed a novel parallel-plate capacitive-force-sensor device based on an electrostrictive terpolymer through drop-on-demand (DoD) inkjet printing technology [20]. Lately in the work of Xiang et al. dedicated to endovascular ablation for varicose vein treatment, an innovative induction-heating device was successfully extruded through a customized ferroelectric composite [21]. During the printing stage, the magnetic particles were perfectly aligned within the polymer matrix along the input field direction, leading to substantial enhancement in the heating effect.

Additive manufacturing (AM)-based 3D printing gives the ability to customize shapes. In addition, it can automate the printing process to save time and feed materials on demand to minimize waste. Numerous printing technologies, including screen printing, aerosol printing, transfer printing, and inkjet printing, have been intensively investigated in the literature [22,23,24,25,26], but sometimes they are expensive and not suitable to high-viscosity materials. Hence, this study examines the extrusion printing technique with further advantages of easy process and reasonable cost. It is obvious that sensor-based printed electronics open new opportunities for next-generation smart monitoring devices. Nonetheless, one of the limitations that are currently preventing wider adaption of AM technology is the lack of understanding of how ink properties affect the AM process and quality of the printed object. Therefore, selection of conductive inks suitable to the target application is of primary importance. In this work, two brands of carbon-based composite ink are chosen and compared in terms of morphological, electrical, and piezoresistive properties. For the sake of simplicity, experimental characterizations are conducted on 2D rectangular substrates made of the same material as 3D suction cups. The purpose here is to select the most adequate ink for the development of SC coating via extrusion-based printing AM. Design optimizations are then thoroughly investigated to enhance the detection area of the sensor coating and minimize the electrode effect. To date, such investigations have rarely reported been in the literature to the best of our knowledge. Accordingly, the findings of this research could pave the way for development of electronic printing methods in the context of SC predictive maintenance.

The paper is organized as follows. After the introduction, Section 2 describes the design strategy of an SC with sensor coating using printing AM. Methods of ink characterizations for 2D and 3D samples are detailed in Section 3, followed by results and discussions of the sensor-coating performance investigated in Section 4. Section 5 presents a preliminary exploration of an improvement in the printing process thanks to the addition of a controllable robot arm with a high degree of freedom (DoF). Finally, Section 6 summarizes the major findings of this paper and provides recommendations for future investigations.

## 2. Design Strategy and Printing Process

### 2.1. Design of Sensor Coating

The suction cup (SC) involved in this study belongs to the VSA series, whose bellows combine the advantages of flat SCs with increased deflection, flexibility, and precision, ideal to handle uneven and rough surfaces. Table 1 shows the features of a Ø78 suction cup made of hydrogenated nitrile butadiene rubber (HNBR) weighing 42 g provided by our industrial partner (name not disclosed for confidential purposes).

For an easier sensor design, the SC is decomposed into three principal zones: top bellows, bottom bellows, and lip (Figure 1a). To obtain the ranges of deformations seen by these structure areas, finite element (FE) modeling was carried out using ANSYS software (previously developed by the industrial partner). As indicated in Figure 1b, the mechanical deformations are mainly located on the bellows of the structure (zones 1 and 2), particularly in the case of the concave surfaces, where the curvatures are the most important. Since the sensors are intended to be printed on the outer surface of the SC, the deformation sensed by the coating will be smaller in the top bellows (~5–8% on convex side) compared to the bottom below (~12–14% on concave side). It is thus essential that the printed sensors are capable of withstanding such mechanical strains under static and dynamic regimes. As expected, the lip (zone 3) does not undergo any deformation, as this part is always fixed, even though the SC is in movement. However, it could be subjected to important stresses pushed by atmospheric pressure on a small seal-contact area. Being the only one interfaced with an object, the lip is considered a strategic part that can suffer from wear and tear. Since atmospheric pressure will always try to equalize itself, air naturally fills in any missing gaps. This pressure pushes against the air outside the SC and forces the lip. If the lip is deteriorated, air can work under the edges of the SC, making the “seal” break and the SC fall off. Consequently, finding a good indicator to assess the lip’s properties in a long-run perspective is mandatory for prevention and maintenance. That is the reason that a circle sensor is also coated on the lip, although this study puts more focus on the bellows’ monitoring. Eventually, the functional coating must adhere well to the HNBR material and be able to operate for several million cycles without deterioration [27]. Piezoresistive technology can meet the application’s needs based on conductive ink [28].

As shown in Figure 2a, the sensor network architecture consists of three circle resistors coated on the critical areas (two bellows and a lip). Vertical lines are designed as electrodes attached to the head of the cup to facilitate the interconnection with electrical spring contacts. The whole sensor network can be modeled as a simple equivalent electrical circuit (Figure 2b), in which each resistive sensor is accompanied by two electrodes. To enhance accurate measurement of the bellows’ deformation, the resistance variations of the sensors must be much greater than those of the electrodes (e.g., a factor of 10). In other words, it is necessary to guarantee that:(1)$$2\;∆R_{i\_electrode}\ll ∆R_{i_{sensor}}$$
where ∆R denotes the resistance variation and the subscript *i* describes the location of the sensor corresponding to *l* (lip), *t* (top), and *b* (bottom). It can be deduced from Figure 2b that $∆R_{i_{global}}$ is the sum of resistance variation by considering the presence of the electrodes. When the condition of Equation (1) is met, $∆R_{i_{global}}$, which can be obtained by experimentation, is expected to be close to the resistance variation of the sensor (i.e., $∆R_{i_{sensor}}\approx \;∆R_{i_{global}})$. Therefore, it is believed that measuring $∆R_{i_{global}}$ leads to an estimation of the deformation of the structure on which is attached the printed coating.

For a given length, the resistance value of the sensors strongly depends on their thickness and width. A compromise should take into account to achieve high sensing performance with large resistance variation. On one hand, the thickness of the printed lines should be as thin as possible to ensure good adhesion between ink and the SC surface. On the other hand, too thin a layer could be fragile and broken under mechanical solicitation, provoking discontinuities in the patterns. Regarding the width of the sensor, it should be small enough to capably detect the deformation at a precise location and large enough to keep the resistance value under the limit imposed by the conditioning electronics. To avoid any problem of parasitic resistance and to ensure good piezoresistive sensitivity, the resistance value of each sensor must not exceed 300 kΩ. Based on these best trade-offs, the patterns are designed with 1 mm width and 20–50 µm thickness. From a practical point of view, 1 mm width is also compatible with the design of 6-electrode lines, which is not a trivial matter. To prevent capacitance parasite, the distance between two adjacent lines should be triple their own width [29]. Furthermore, the dimension of the 6-electrode line terminal must fit with the electrical connector provided by our industrial partner.

### 2.2. Ink Selection

Various conductive inks are now commercially available, but few adhere to HNBR and are sufficiently mechanically resistant to withstand a huge deformation. To reach the target application, the selected inks must fulfill the following criteria:good adhesion to the SC’s materials (HNBR), particularly for the curved-surface bellows;excellent stretchability to resist high deformation (~20%) under static and dynamic conditions;adequate electrical conductivity to counterbalance the electrode effect;high viscosity to achieve shape fidelity (as no thermal treatment is performed throughout the printing process);stable electrical behavior under aging test.

After testing several conductive inks, only two products met the first criterion, where the sensor coatings adhered to the SC surface after printing. The first one, supplied by Creative Materials (CM, 128-07 product), possesses lower viscosity, but shape fidelity and adherence to the HNBR material are also attained. The second one exhibits very high viscosity (~57,000 cps) compared to those used in classical screen printing [30,31,32]. For confidential purposes, the name of the second ink’s supplier is not shown. Both inks were made of polymer-based carbon particles whose chemical composition was analyzed later using microscopy (Section 4.1). Table 2 provides essential properties of the two inks that are necessary for the printing and curing process.

For the sake of simplicity, characterization tests of the printed coating comprising morphological analyses, electrical conductivity, and piezoresistive behavior are conducted on 2D samples (dimension 120 × 20 mm). These samples, provided by the industrial partner, are made of HNBR material, i.e., identical to the one of the 3D suction cups.

### 2.3. Printing Process

Figure 3 illustrates the printing setup conducted on a 2D sample using the Hyrel 3D System (ref. 30 M), which consists of a glass platform and a modular head extruder (cold flow at room temperature) capable of translating in three directions. To efficiently extrude high-viscosity ink, a powerful printhead extruder (VOL-25) was chosen, together with an adequate stainless-steel nozzle tip whose inner diameter equals 1 mm. The syringe pump extruder has a standard ink reservoir of 25 mL containing fluids that were expected to be printed into desired patterns. To perform the desired printing trajectories, relevant parameters affecting the printing quality, such as layer height, ink flow rate, line density, pressure control, etc., are carefully tuned in the Slic3r software. After several adjustments, the flow rate and pressure control were respectively set at 300 pulse/µL and 0.8 (1 is the default). Finally, a simple printed-line resistor was coated on a 2D substrate using either the first or the second ink (see Figure 3). At each extremity of the resistor, a squared coating was designed to be used as an electrical connection necessary for practical characterizations.

It is obvious that printing a straight-line resistor on a 2D substrate is not the same task as printing three circle-shaped lines on a 3D cup, which is a complex shape with curved bellows. Another 3D printer model (Hydra 16A), comprising a larger workspace platform with the addition of a motorized support, was used to make the task feasible. As depicted in Figure 4, the sample’s support can be rotated in a full range (0–360°) around the Y-axis through the transmission of shaft B, and 0–180° around the X-axis thanks to shaft A. Based on this device, the coordinates of the SC in the X- and Y-axes can be adjusted by setting the rotation of these two shafts (with precision of 0.1°). The control of the motors is given by the Repetrel program where the CAD model is imported and converted into Gcode language. The choice of the printhead, nozzle tip, and syringe pump extruder are the same as for the 2D-sample protocol. Also, the extruder system has a 3 degree-of-freedom (3-DoF) translation. Perfect synchronization between the support movement (2 DoF in rotation) and the extruder movement (3 DoF in translation) is undoubtedly a key issue of the fabrication success.

To achieve the desired patterns of the SC and to improve the ink adhesion, the quantity of the output ink (adjusted by extrusion speed) and the distance between the nozzle needle tip and the substrate surface during the operation are of primary importance. This distance should be as small as possible, about 0.2–0.3 mm, so that ink is injected with sufficient force to be quickly adhered on the substrate. Accordingly, the 3D printing process requires high precision of the SC position, the distance between the needle tip and the coating, as well as the synchronization between the motors’ speed and the amount of the output ink. To some extent, these parameters substantially affect the pattern’s shape fidelity and thus the print quality. Appendix A shows the whole AM procedure, which allows to observe full conductive coating printed on a SC surface.

Once the printing process is complete, the 2D and 3D samples are placed into an oven (Memmert V0 400 drying oven) under an adequate thermal condition to obtain cross-linking polymerization of the inks. Following the recommendations given by the ink’s suppliers, the coated samples are cured at 120 °C for 30 min for the second ink, and 175 °C for 10 min for the first ink.

## 3. Characterization Methods

### 3.1. Morphological Characterization

To visualize the surface state of the sensor coatings, morphological characterization via scanning electron microscopy (SEM) was performed. Observation tests were conducted on the top surface of 2D samples using compact SEM equipment (FlexSEM 1000II, Hitachi High-Tech, Tokyo, Japan). The instrumentation is also equipped with an EDS (energy-dispersive spectroscopy) system, allowing for the chemical analysis of inks’ features being observed on an SEM monitor. Eventually, signals produced in the SEM/EDS system include secondary and backscattered electrons that are used in image forming for morphological analysis, together with X-rays that are used for identification and quantification of chemicals present at detectable concentrations. In general, EDS can detect major elements with concentrations higher than 10 wt% (major), and minor concentrations between 1 and 10 wt%. The detection limit for bulk materials is 0.1 wt%; therefore, EDS cannot detect trace elements below 0.1 wt%. It is well known that the detection limit in EDS depends on sample surface conditions: the smoother the surface, the lower the detection limit [33]. In practice, the coating surface is considered smooth enough to make EDS experiments reliable, where surface treatment of the printed coating is not necessary. To verify the accuracy of the chemical composition analyses, EDS was performed on several selected microareas of the sample’s surface.

### 3.2. Electrical Characterization

Electrical characterization of the composite inks is carried out based on the determination of their resistance (denoted R). This allows for estimation of the material’s conductivity (denoted σ) in the following expression [34]:(2)$$\sigma =\frac{l}{R\times S}$$
where S and l, respectively, denote the surface and the length of the printed coating. S is determined as S=e×w, where e and w are correspondingly the thickness and the width of the coating (see Figure 5a).

There are many methods for determining the resistivity of a material, but the technique may vary depending upon the type of material, magnitude of the resistance, shape, and thickness of the material. One of the most common ways of measuring the resistivity of thin, flat materials like semiconductors or conductive coatings is to use a two-point probe method. Such a technique involves bringing two probes into contact with a material of unknown resistance. A DC current is applied between these two probes, and a voltmeter measures the voltage difference between them. The resistivity is computed from geometric factors, the source current, and the voltage measurement. For making a direct resistance measurement, the Ohm-meter (RM_804) used for this test includes a DC current source, a sensitive voltmeter, and two alligator-clip probes clamped on the two ends of printed coating (see Figure 5b).

### 3.3. Piezoresistive Characterization

#### 3.3.1. D Rectangular-Shaped Sample

Experimental monotonic tensile testing based on ASTM D638 standard [35] is applied to assess the piezoresistive properties of the printed sensors coated on the HNBR elastic substrate. Figure 6 shows the typical tensile configuration using a Shimadzu (AGS-X model) machine where the 2D specimen is firmly clamped by two pneumatic-actuated grips: the top grip can be freely moved in translation (via a controlled motor) while the bottom grip is fixed. By controlling the distance between the two grips, it is possible to impose a desired displacement (denoted x) on the sample. The resistance variation of the printed sensor is thus evaluated as a function of the amplitude and the rate (i.e., speed) of the applied displacement, which is designed as a periodic triangle or trapezoidal waveform.

Knowing the speed of the tensile test (denoted v where v=dx/dt), it is possible to deduce the strain rate, given by:(3)$$\dot{\varepsilon }\left(t\right)=\frac{d\varepsilon }{dt}=\frac{1}{L_{0}}\frac{dx}{dt}=\frac{v\left(t\right)}{L_{0}}$$

Here, ε is the strain (or deformation) defined as the ratio between the displacement (x) and the initial length of the specimen (i.e., $L_{0}=60\;\mathrm{mm}$). In experiments, 2D samples are mechanically solicited by somewhat slow movement where the maximum speed is set at 4 mm/s, which corresponds to a strain rate of ~0.067 $s^{-1}$.

Finally, the output resistance and the input displacement signals are simultaneously acquired and recorded in real time with a Sirius 8XSGT card interfaced with DEWE software. Subsequent data analysis was performed with MATLAB and Excel.

#### 3.3.2. D Suction Cup

Figure 7a illustrates a specific test bench developed by our industrial partner, which allows for the characterization of printed sensors in real conditions. Thanks to an automatically controlled vacuum pump system, it is possible to actuate the suction cup (SC) under two operating modes: (1) compression—the SC is subjected to vacuum when being pressed against a tested surface; and (2) release—the vacuum is interrupted, making the SC return to its original shape. To some extent, variation in the SC’s shape leads to change in the resistance of the printed sensors. For comparison, the experimental setup enables testing two SCs simultaneously: one of them is performed on a standard flat surface, while a specific configuration (i.e., incline, angle, or sphere surface) is used for the other (see Figure 7b). The SC is intentionally designed to be used on several types of objects with different surfaces. The test bench is powered by a 24 V DC generator. The resistance measurement is acquired by an acquisition card interfaced with homemade software, allowing for monitoring and recording the signals in real time. The multichannel box, also connected to acquisition card and computer via homemade software, is used to generate an input waveform to drive the vacuum pump system. Parameters relating to the number and duration of the input cycles can be modified online from the computer.

## 4. Results and Discussion

### 4.1. Morphological Properties

Figure 8 shows the SEM images of the first ink (a-b-c) and the second ink (d-e-f) made with three magnifications (×75, ×1000, and ×2000, respectively). As observed, both of them lead to homogeneous printed surfaces, despite some tiny defects appearing on the coating layer. The second ink somehow exhibits a smoother microsurface than its counterpart (Figure 8e,f vs. Figure 8b,c), which possibly originates from differences in the chemical compositions and the particle size of these two inks. The EDS analysis shown in Figure 9 allows to partially confirm this hypothesis. The percentage range of the elements in the composite mass is visualized in the adequate accelerating voltage range and counted in seconds per electron volt (cps/eV). It has been pointed out that both inks are made of major elements like carbon with high concentration (>80 wt%), but other minor elements are not the same (e.g., 9.3 wt% of Cl for the first ink against 11.5 wt% of O for the second ink, and so on), confirming their dissimilarity in the chemical formulation.

### 4.2. Electrical Properties

Based on the measurement of the resistance and dimensions of the printed lines made of the two selected inks (Table 3) on a 2D substrate (Figure 5a), it is possible to infer their electrical conductivity (σ): 90 S/m and 140 S/m, respectively. These values are somehow smaller than those given in the inks’ datasheet. It is worth noting that the measure uncertainties strongly depend on the uniformity of the coating’s thickness as well as the surface contact between the probes and the sample. Although the two inks exhibit low value of σ, their electrical properties are revealed to fit with our target application. The resistance value of each sensor coating printed on a 3D suction cup must not exceed 300 kΩ (either in static or dynamic regime), which is the value limit imposed by the acquisition software.

### 4.3. Piezoresistive Properties

#### 4.3.1. D Rectangular-Shaped Sample

(a)First ink

Tensile tests were conducted on a 2D specimen coated with the first ink, i.e., subjected to a maximum deformation (or strain) of 10%. As demonstrated in Figure 10a, the resistance and the applied input strain are perfectly in phase, reflecting good correlation between the electrical and mechanical properties of the ink. Interestingly, when the strain falls from the maximum (10%) to the minimum (initial state at 0), the resistance is held in a few seconds. HNBR material is an elastomer that tends to shrink when being deformed [36]. This shrinkage needs time to return to its initial state [37]. A period of 20 s applied in this test is probably too fast for the shrinkage, which causes the nonlinearity of the resistance signal from 2% down to 0. For the rising phase of the strain from 0 to 10%, the resistance exhibits a small overshoot in a transient regime and then gradually decreases to attain a steady state.

Figure 10b confirms a linear relationship between the electrical resistance (*R*) and the mechanical strain, i.e., accompanied by somewhat hysteretic behavior linked to relaxation behavior of the elastomer. The linearity can be explained according to Equation (1), where the specimen’s length affected by the tensile strength is proportional to *R* and so is the deformation, assuming that the section of the printed line is unchanged and variation in the resistance is mainly due to variation in length. During the rising and falling phases of the deformation (*ε*), the ink exhibits the same piezoresistive coefficient of around ΔR/ε=8.6 kΩ/%, leading to a determination of the gauge factor (*GF*) according to the following expression:(4)$$GF=\frac{∆R}{R_{0}\times \varepsilon }$$
where *R* is the resistance of the undeformed gauge (~50 kΩ); and ∆*R* is the change in resistance caused by strain *ε*. The *GF* is then deduced to be equal to approximately 17.3, which is consistent with the value found in the literature for plain-woven carbon fiber reinforced polymer (CFRP) [38]. Noted that most standard commercial formulations have positive piezoresistance with gauge factor in a range between 2 to 4.

Besides the deformation, the tensile rate (or speed) is also a relevant factor that can influence the resistance change (∆*R*). The geometry of the fillers within the polymer matrix decides the reaction rate of the ink under a given deformation [39]. As displayed in Figure 10c, ∆*R* is somehow stable for different speeds from 0 to 4 mm/s, where discrepancy is less than 8%. Considering uncertainties of measurements, it is reasonable to discard this discrepancy and consider that ∆*R* is unchanged as a function of the tensile rate. In conclusion, the sensor printed with the first ink exhibits excellent piezoresistive response with good stability, high sensitivity and linearity.

(b) Second ink

First, for the tensile test, a 2D specimen coated with the second ink is subjected to a constant speed (i.e., 1 mm/s) and varied strain. Figure 11a,b illustrate the resistance vs. the displacement with different amplitude of 4 mm and 10 mm (i.e., equivalent to a strain of ~7% and 17), respectively. As can be seen from the blue curve, the resistance (*R*) is not linear regarding the displacement (denoted *x*). Furthermore, the hysteresis behavior allows to infer that *R* does not only depend on *x* but also on its derivation dx/dt (so-called speed). An analytical model of the electrical resistance can be given by:(5)$$R(x)=R_{0}(x)+f(x)\frac{dx}{dt}$$
where *R*_0_ is the fifth-order polynomial model dedicated to the resistance trend (the red curve displayed in Figure 11a,b) determined by the curve-fitting tool of MATLAB, and f(x) denotes the speed factor, which was estimated based on the discrete interpolation of  R(x) function regarding the displacement. This method involves the construction of data points for the f(x) function so that the interpolated signal (yellow curve) built from Equation (5) is fairly close to the original one (blue curve). Figure 11d shows the results of the speed factor estimated with different displacement magnitude (denoted *A*) from 4 mm to 10 mm. It has been highlighted that under a constant speed, the f(x) factor is almost similar whatever the value of *A*, meaning that the variation in *A* only leads to the change in the resistance trend (*R*0), but not in the hysteresis behavior (second term of Equation (5)). Interestingly, the hysteresis is maximum at the two extremities of the displacement (equal to 0 and *A*) and minimum at the displacement near to A/2 (Figure 11d). This observation is somehow related to the nonlinearity of the resistance, which is accentuated at the extremities and almost linear around the center of the displacement’s interval.

Unlike the first ink that has positive piezoresistance in which electrical resistance increases proportionally with the increased mechanical strain, the second ink exhibits a negative behavior ($R_{0}<0)$. Similar results have been reported in the works of Wang et al. [40] and Mei et al. [41] for the CFRP using either a two or four probe-wire method. Both argued that the positive piezoresistance that was previously reported by many researchers was caused by increased electrical resistance in electrodes, not from the changes in electrical resistance of CFRP. Todoroki and Yoshida conducted tests with similar techniques to [40,41] and produced negative piezoresistance [42]. However, they contested that the negative piezoresistance occurred due to poor surface treatment prior to the electrode attachment, which is contrary to fine surface treatment. The latest study was conducted by Jeon et al. to find the effect of contact resistance between CFRP and electrodes to the piezoresistive behavior [43]. They also reported that the roughness of the composite surface attached to the electrodes affects the test results. Thus, these factors need to be considered in future testing and application.

Another test is set with a constant displacement of 6 mm (i.e., corresponding to 10% strain) and varied speed with amplitude of 0.5 mm/s and 2 mm/s, as shown in Figure 12a,b. To some extent, the resistance change (∆*R*) somewhat increases with the speed value. Figure 12c represents the time evolution of the resistance and the displacement, in which these two signals are completely out of phase (i.e., the phase shift equals 180°). This confirms the negative piezoresistive response as described above, suggesting that the geometric factor has a fundamental impact on the resistance value. Based on the nonlinear model of Equation (5), it is possible to estimate the f(x) speed factor in such a way that the estimated resistance (yellow curve in Figure 12a,b) perfectly fits the measured one (blue curve). It is clearly shown in Figure 12d that the f(x) factor is velocity-dependent and tends to decrease with an increase in the speed’s amplitude. Exceptionally, very high speed makes f(x) tend to 0 and the resistance becomes anhysteretic, whatever the applied displacement [44].

Accordingly, regarding the above characterization, it is clear that the first ink is more suitable to be involved in the development of sensor thanks to its linear and positive piezoresistive response. The second ink, despite its nonlinear and negative piezoresistance that would make calibration and analyses complicated, exhibits better adhesion to HNBR material after extended mechanical solicitation. Since adhesion is a critical issue in additive manufacturing (AM), especially for curved surfaces (e.g., bellows), the second ink is finally chosen to be coated on a 3D suction cup.

#### 4.3.2. D Suction Cup

Figure 13a illustrates the experimental setup described in Section 3.3.2 that allows for an assessment of the resistance evolution of two suction cups (called SC1 and SC2) simultaneously, one tested on a flat surface and the other on the surface with different configurations (flat, incline, sphere, and angle). Both SCs were printed with three circle-shaped sensors using the second ink (Figure 13b), but observations were mainly focused on the top and the bottom bellows where the deformation was supposed to be important, and so was the resistance change.

Figure 13c displays FFT (fast Fourier transform) analysis of both SCs’ resistance measured on the two bellows’ sensors for different surface configurations. Actually, FFT investigation is generally used to analyze the frequency spectrum of a signal, in which only amplitude variation is getting involved, regardless of its original state. As expected, the resistance magnitude of the concave bottom bellows is higher than that of the convex top bellows, which agrees with the strain distribution shown on the finite element (FE) modeling (Figure 1b). Investigation performed on the 2D substrate coated with a printed-line sensor showed a resistance change of approximately 25 kΩ under a 10% deformation (Figure 12c). To simplify the estimation of the SC’s strain, a quasilinear relationship is considered. As demonstrated on Figure 12d, hysteresis behavior (i.e., image of the speed factor) could be negligible under a fast dynamic control of the vacuum pump system. In the case of SC1 tested on the flat surface, the resistance variation of the top and the bottom sensors, respectively, equals 43 kΩ and 17 kΩ, leading to a strain prediction of 18% and 7%. For SC2, these values are smaller, i.e., 15% and 4% correspondingly. To some extent, these findings somehow relate to the simulation result shown in Figure 1b, even though we used the hypothesis based on a linear model, which is far from reality.

Among different surface configurations (cf. Figure 13c), the flat surface leads to the highest resistance value, followed by the spherical and inclined surfaces, while the angled surface makes the lowest contribution. On one hand, the applied compression force is perpendicular to the flat surface, giving rise to maximum deformation that is symmetrical with respect to the central axis. On the other hand, the force is no longer perpendicular to the other types of surfaces, resulting in smaller deformation of the SCs. Particularly for the angled and spherical surfaces, the inner vacuum of the SC is somewhat filled by air through tiny gaps between the lip and the curved surface, provoking lower vacuum pressure and thus smaller resistance change.

Aging tests were carried out to verify whether or not the adhesion between the carbon-based-composite ink and the SC surface can endure extended mechanical solicitation. Figure 14 displays the resistance measurement of the top and bottom sensors coated on two SCs’ bellows for almost 1 h. For all cases, the resistance slowly decreases during the transient regime occurring for the first 500 s, which relates to the mechanical relaxation of the HNBR’s material. After around 100 cycles, the resistance magnitude becomes stable and reaches the steady regime. Throughout 1 h testing, the measured value remains unchanged, confirming high reliability of the developed sensing device based on additive manufacturing (AM) technique. Inset graphs reveals a square waveform of the resistance evolution in the time domain, which is in agreement with the FFT spectrum of Figure 13c in the frequency domain. A square wave consists of a fundamental sine wave (of the same frequency as the square wave) and odd harmonics of the fundamental whose amplitude decreases and proportional to 1/N (N denotes the harmonic, N=1, 3, 5). The fundamental frequency was found to be 0.25 Hz, corresponding to the input waveform rate used to control the vacuum pump system. During a period of 4 s, compression and release phases are getting involved. First, the SC is pressed against a surface to create a vacuum inside that sucks the bellows, making the strain increase (in absolute value) and so the resistance variation. Second, the atmospheric air fills the released SC, which tends to its initial shape, thus reducing the resistance value.

To assess the impact of the electrodes, two samples were conducted: one consisted of full sensor coating while the other was made with electrode lines only (cf. Figure 15a). A comparison of these samples was performed via analysis on the FFT peak value (i.e., deduced from the fundamental harmonic) with the four surface configurations (cf. Figure 15b). A surprising result was found at the lip’s measurement, where the resistance change of the electrode coating was a little bit superior to the one of the full coating, regardless of which configuration was chosen. This maybe came from inevitable measurement uncertainties due to printing quality, homogeneous ink, unperfect setup, and so on. More thorough analysis dedicated to the lip’s wear monitoring will be involved in our future investigations. Regarding the top and the bottom bellows (see Figure 15b), the full sensor coating exhibits clearly higher FFT peak value (around factor 2) than the electrode coating, whatever the surface configuration. These results clearly confirm the influence of the electrodes, which would be minimized in the design optimization so as to improve accuracy of the resistance measurement.

Therefore, an alternative solution was considered, as illustrated in Figure 16a: the SC was designed with hybrid coatings where the electrodes were printed with another silver ink (Creative Material, ref. 127.07), with no change for the circle-shaped sensors. The selected silver ink has the advantage of excellent electrical conductivity, making its resistance negligible compared to the one of the printed sensor. As expected, in Figure 16b, the hybrid SC leads to substantially smaller resistance with respect to the full coating printed with the second carbon ink. Regarding the resistance change, the FFT peak value displayed in Figure 16c is reduced for both sensors, particularly in the case of the top bellows. Such results are very encouraging, confirming high benefit of the hybrid design in which the influence of the electrodes could be ignored. Unfortunately, the hybrid prototype in reality is not robust enough to go through the aging test under a great deal of stress, because of poor adhesion of the silver ink.

The last design involved using only the second carbon ink, but increasing the length of the two sensors so their resistance variation was sufficiently superior to the electrodes’ one. As illustrated in Figure 17a,b, each bellows was printed with three circle-shaped resistors, which can be connected either in series or in parallel. The serial-connection sample was first fabricated, then transformed to the parallel counterpart by adding short printed lines (illustrated as red lines for an easier visualization). Such a design allowed to enhance the detection area of the sensitive elements, with the aim of monitoring the global state of the bellows. In the former sensor design based on a one-circle resistor, the printed lines had a 1 mm narrow width located at the center of the bellows. To some extent, this position did not cover all relevant areas of the bellows where the deformation was supposed to be meaningful. As a result, a new design involving a three-circle resistor seemed to be a good alternative for an enhancement of the detection area.

Table 4 shows the absolute resistance variation (∆R, where $\Delta R=R_{max}-R_{min}$) of the three SCs shown on Figure 16 and Figure 17. Logically, the one coated with three-circle serial resistors gives rise to the highest value of ∆R, whereas the parallel counterpart results in the smallest value. Whatever the coating’s design, ∆*R* of the top bellows is significantly small compared to that of the bottom bellows, which agrees with the simulation and empirical results investigated above. To further highlight the comparison of the three designs, another indicator—the relative resistance variation (${∆}_{m})$—is given as follows:(6)$${∆}_{m}=\frac{R_{max}-R_{min}}{R_{max}+R_{min}}=\frac{∆R}{R_{m}}\times 100\%$$
where $R_{m}$ denotes the mean resistance that is computed by $R_{m}=R_{max}+R_{min}$.

Regarding the results shown on Table 4, the one-circle hybrid coating gives rise to the maximum value of ${∆}_{m}$ performed on the bottom bellows. The three-circle parallel coating, on the contrary, induces the minimum value. It can be therefore suggested that the center of the bottom bellows is the most deformed area, and thus adding printed circles allows for an increase in the detection area but does not enhance the relative resistance variation (${∆}_{m}$). Concerning the top bellows, however, all designs exhibit moderate ${∆}_{m}$ value because of the smaller strain detection. Interestingly, the parallel coating gives rise to somewhat higher variation with respect to the other (i.e., 7% vs. 5%). Finally, the three-circle serial coating exhibits comparable performance to the one-circle hybrid coating. Although three-circle patterns are more complex, this design makes it possible to employ only one type of ink. Inversely, the hybrid coating uses a supplementary silver ink whose adhesion and stretchability are still critical issues in 3D printing AM.

To sum up, the samples printed on a 2D or 3D structure, even with the same inks, somehow exhibit different performances. Among them, the main highlights can be listed as below:Adhesion of the printed lines are better in the 2D planar specimen than in the 3D curved structure, regardless of which ink is chosen.The second ink is well adhered in both 2D and 3D samples under extended mechanical solicitations, while the first ink could only withstand with the 2D structure. Actually, dynamic test using the vacuum pump system carried on the 3D structure is faster than the tensile test performed on the 2D specimen.The 3D cup with such complex curves makes the ink selection challenging. Among several tested inks, only one could meet our expectations in terms of technical feasibility. However, this ink exhibits nonlinear and negative piezoresistance, leading to complex sensor calibration and data analyses, to some extent.

## 5. Future Developments for Printing-Process Enhancement

To better improve sensor sensitivity as well as the detection, an innovative design using crenelated patterns coated on the two bellows is intended to be investigated in future work (see Figure 18). Each printed crenelation should be built in such a way that its resistance does not exceed 300 kΩ so as not to alter the sensitivity of the sensor (relating to strain gauge factor). Unfortunately, the limitation of our current printer does not allow us to achieve such a complex design. Development involved in the robot control together with enhancement of ink stretchability is under exploration to fit with these complex patterns, especially when the substrates are extremely curved, as in the case of the concave and convex bellows.

To completely automate the printing process, a six-axis robot arm control (TX2-60, Stäubli Controller: CS9) developed by another company (name not disclosed) was initially investigated, as shown in Figure 19. The 30M Hyrel extruder drives the printhead nozzle that is kept fixed, while the robot arm carries the SC and performs movements in 6 DoFs. Including an automatically controlled robot arm into AM makes the printing stages more manageable and less time-consuming, as the robot can perform more complex and precise trajectories than the printhead itself. Our preliminary results have demonstrated the feasibility of this approach, which have provided a real breakthrough in the field of sensor coating-based 3D printing technology.

## 6. Conclusions

In this study, we propose an innovative method for condition monitoring a suction cup (SC) based on the resistance measurement of printed conductive coatings. We focused on the piezoresistive behavior of the two circle-shaped sensors coated on the SC bellows, where deformation was revealed to be considerable during vacuum operation regarding the simulation result. On the other hand, deformation of the SC lip was demonstrated to be negligible, thus not being considered in this work. However, a sensor coating was still printed on the lip’s surface for future work related to mechanical failure monitoring in the long term. The lip is not subjected to high strain variation during the cup’s action, but is exposed to important static pressure as being the only part that has a contact with the subject. Success of the printed sensor coatings is essentially impacted by the properties of conductive ink, which must ensure good adhesion to the HNBR material and be capable of undergoing an important deformation (~20%) throughout several compression–release cycles.

Although conductive inks have been intensively explored in 3D printing technology, their piezoresistive behavior still needs to be further clarified. For the sake of simplicity, experimental characterizations were first conducted on 2D substrate-based HNBR material. A tensile unidirectional method was performed to figure out the relationship between electrical resistance and mechanical strains, through which the piezoresistive behavior of the two carbon-based-composite inks was clarified. The first ink exhibited a perfect linear characteristic but poor adhesion to the HNBR substrate under high dynamic solicitation. The second ink fitted better to the target field because of its excellent adhesion and stretchability, despite its nonlinear and negative piezoresistance. Therefore, only the second ink was selected to be printed on 3D SCs conducted on different surface configurations (flat, incline, sphere, angle). The initial design consisted of one-circle-shaped sensors coated on the two bellows and the lip demonstrated the reliability of the selected ink (i.e., the only “survivor” after the aging test). With the aim of enhancing the surface detection of sensitive elements as well as minimizing the effect of electrodes, design optimizations involved the development of hybrid coatings and three-circle sensors connected in series or parallel. To achieve more complex and higher-performance designs, our study oriented to automatic 3D printing using a controllable six-axis robot arm. Preliminary results were encouraging for upcoming exploration in the failure detection and predictive maintenance of SC. Another aspect of this research focuses on investigating new conductive inks with improved features in terms of viscosity, linear piezoresistive response, high stretchability and adhesion. The relationship of such parameters needs to be revealed for the future implementation and development of condition-monitoring technology.

## Figures and Tables

**Figure 1 micromachines-13-01606-f001:**
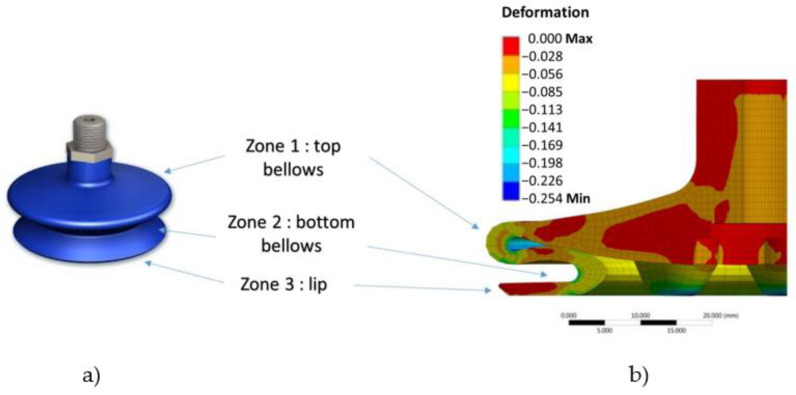
Decomposition of three zones that need to be monitored: (**a**) An SC provided by the industrial partner; and (**b**) mechanical deformation of SC based on FEM.

**Figure 2 micromachines-13-01606-f002:**
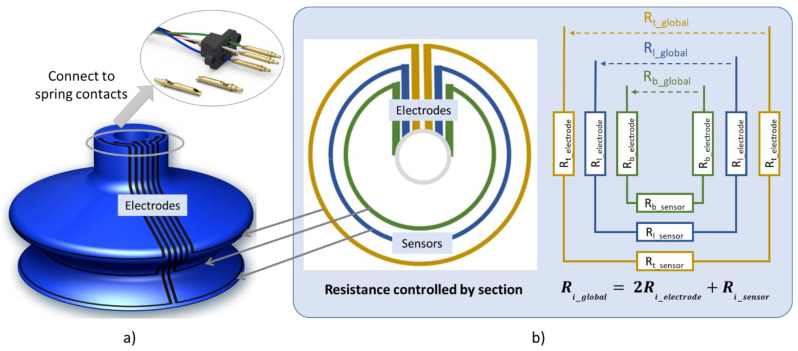
Design of sensor coating: (**a**) prototype consisting of 3 printed circle-shaped sensors coated on the lip, top, and bottom bellows; (**b**) equivalent resistance defined by different zones.

**Figure 3 micromachines-13-01606-f003:**
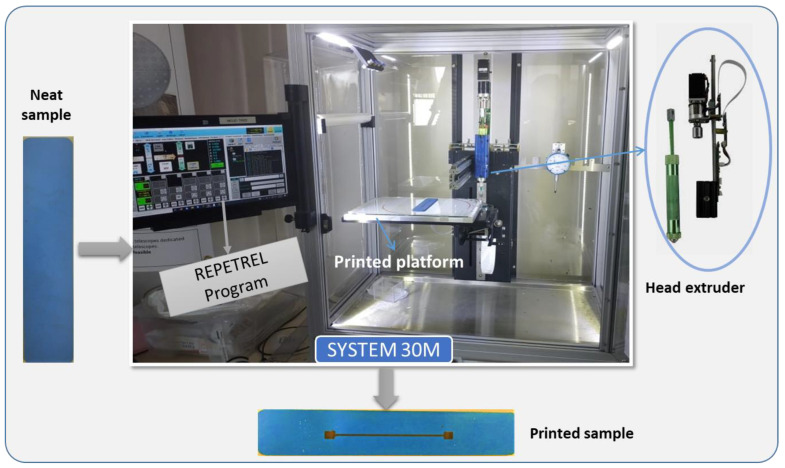
The printing process used to make one printed-line coating on a 2D sample.

**Figure 4 micromachines-13-01606-f004:**
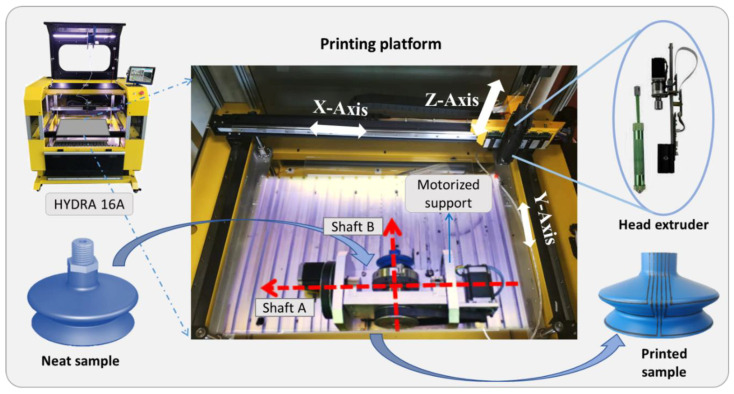
The printing process used to coat 3 resistive sensors on a 3D suction cup.

**Figure 5 micromachines-13-01606-f005:**
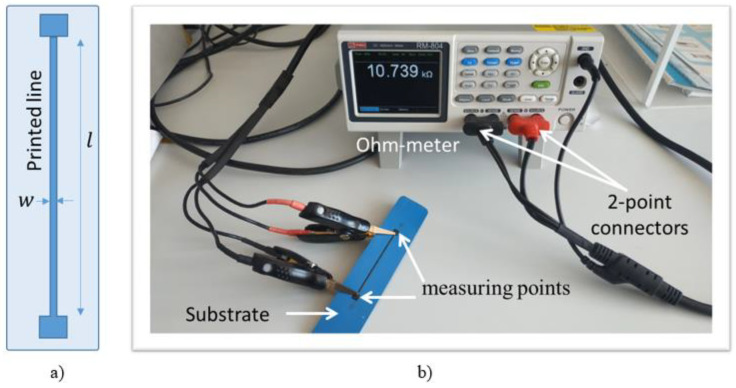
(**a**) Conductive coating printed on a 2D substrate; (**b**) experimental setup used for measurement of electrical resistance.

**Figure 6 micromachines-13-01606-f006:**
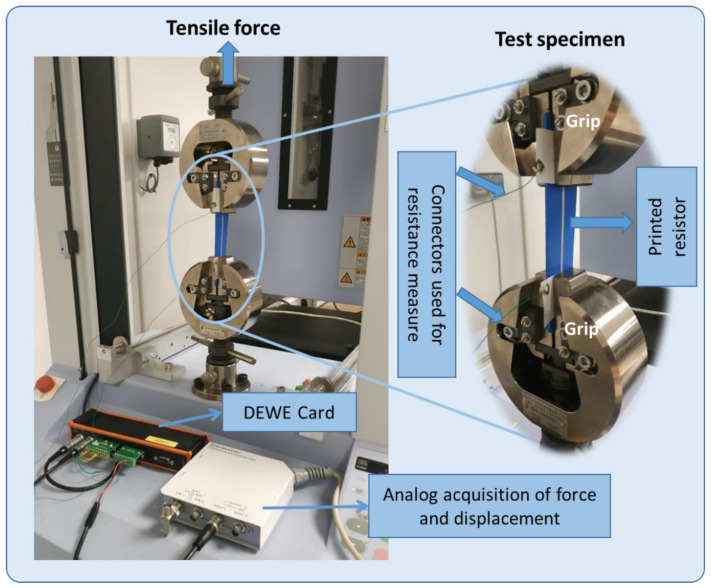
Experimental setup used for a piezoresistive characterization of 2D coated samples.

**Figure 7 micromachines-13-01606-f007:**
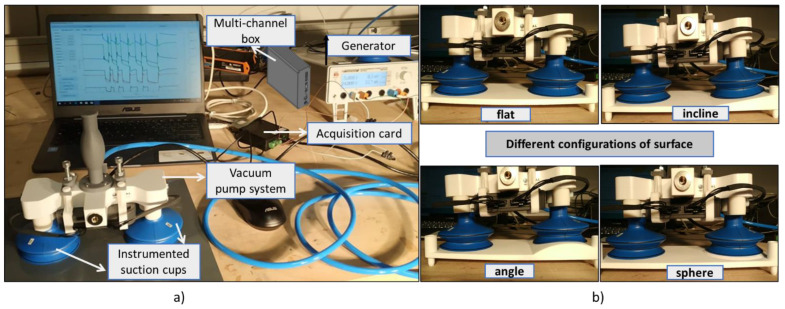
Experimental setup used for piezoresistive characterization of 3D coated suction cups: (**a**) specific test bench with acquisition system; and (**b**) four configurations with flat, inclined, angular, and spherical surfaces.

**Figure 8 micromachines-13-01606-f008:**
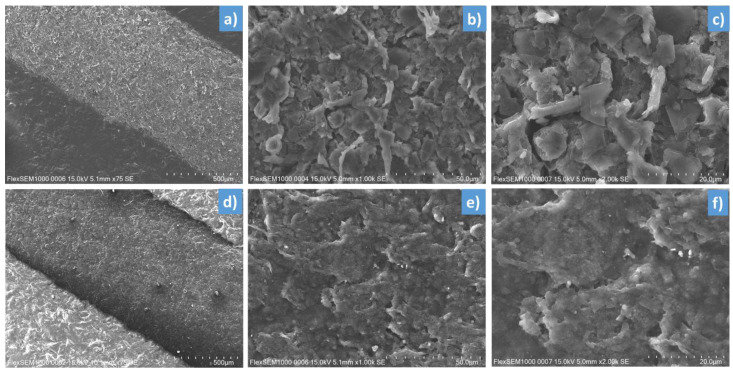
SEM image of 2D samples made of (**a**–**c**) the first ink, and (**d**–**f**) the second ink at magnifications of ×75, ×1000, and ×2000 (from the left-hand side to the right-hand side).

**Figure 9 micromachines-13-01606-f009:**
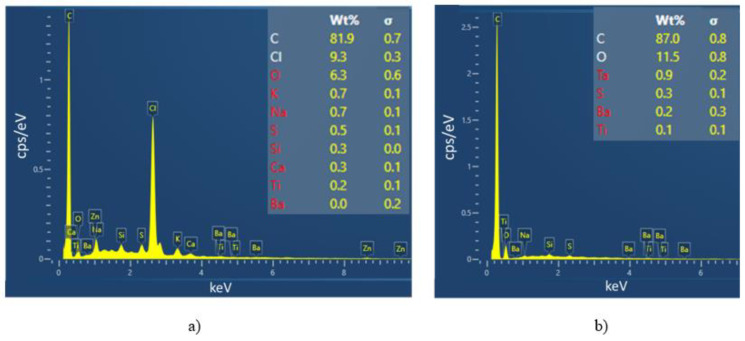
Elemental distribution maps based on SEM-EDS spectrum analysis for (**a**) the first ink, and (**b**) the second ink.

**Figure 10 micromachines-13-01606-f010:**
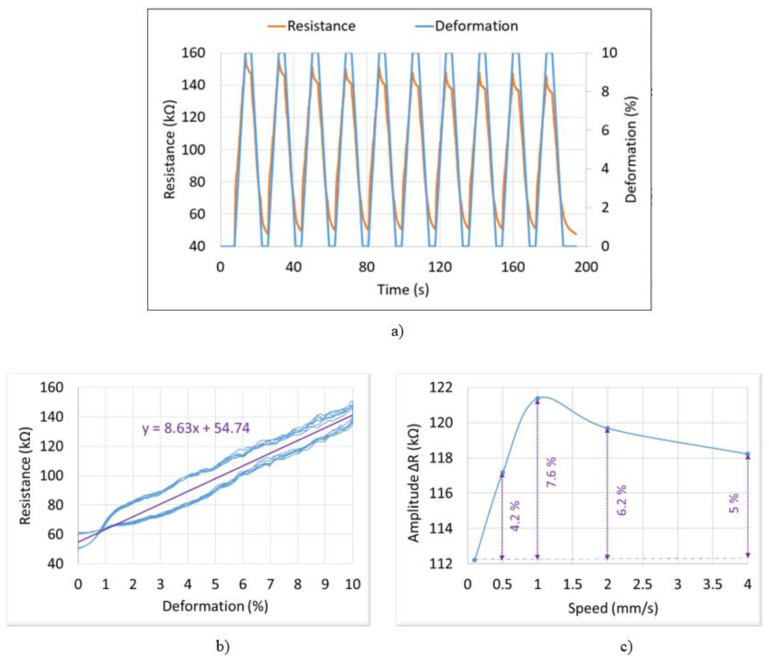
Behavior of the first ink based on tensile tests: (**a**) time evolution of resistance (red) and deformation (blue), (**b**) resistance vs. deformation, (**c**) resistance variation (∆*R*) vs. speed at a 10% deformation.

**Figure 11 micromachines-13-01606-f011:**
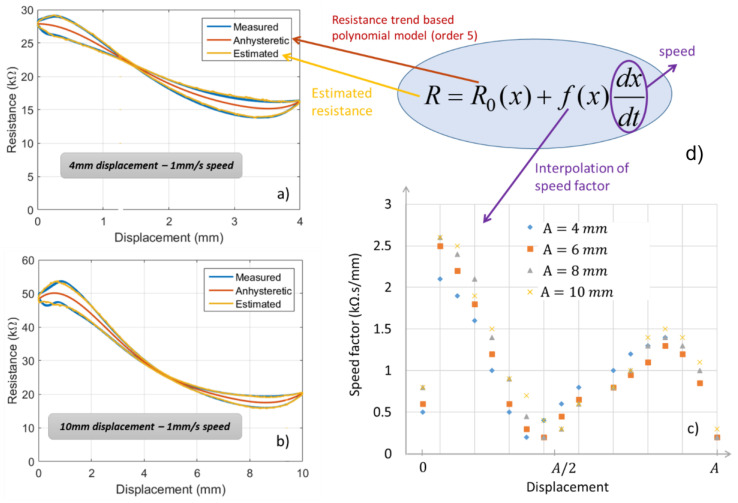
Behavior of the second ink subjected to different mechanical displacements (with speed constant of 1 mm/s): (**a**,**b**) resistance vs. displacement with amplitude of 4 mm and 10 mm correspondingly, (**c**) formula of the estimated resistance, (**d**) interpolation of the speed factor.

**Figure 12 micromachines-13-01606-f012:**
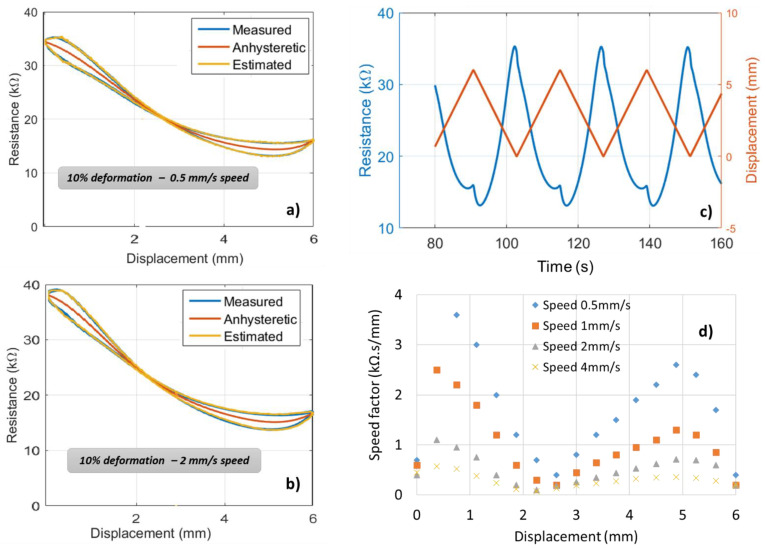
Second ink behavior subjected to a 10% deformation with different dynamics (speeds): (**a**,**b**) resistance vs. displacement under a speed of 0.5 mm/s and 2 mm/s correspondingly, (**c**) time evolution of resistance and displacement, (**d**) interpolation of the speed factor.

**Figure 13 micromachines-13-01606-f013:**
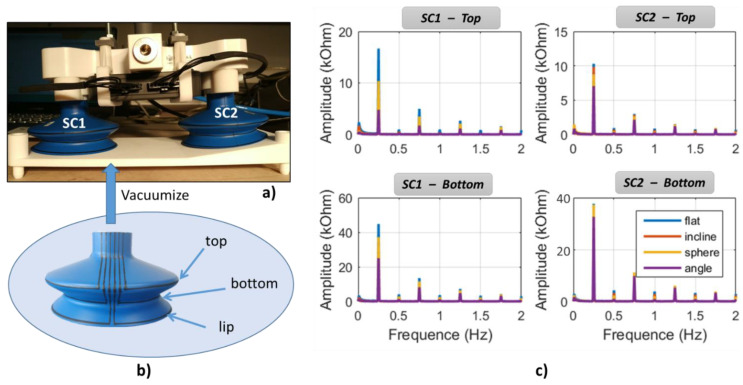
Resistance variation measured on different configurations: (**a**) setup of 2 suction cups (SC1 et SC2), (**b**) three sensors coated on the samples, (**c**) resistance variation of the top and bottom bellows measured on different surfaces for SC1 et SC2.

**Figure 14 micromachines-13-01606-f014:**
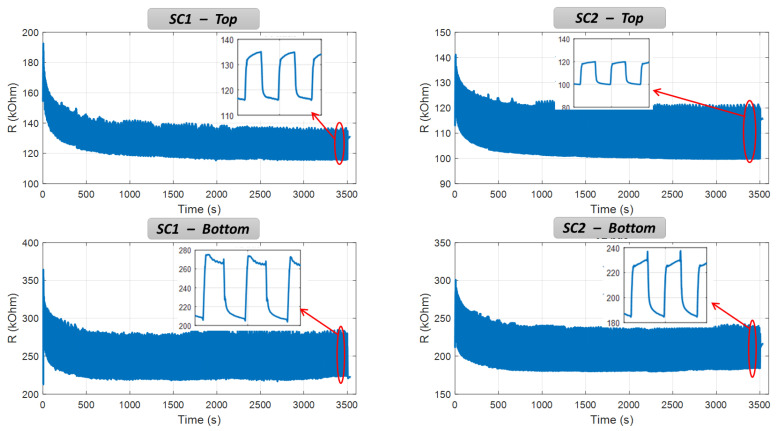
Aging test based on resistance measurement of the top and bottom coatings for SC1 and SC2. Inset graphs zoom in on the last portion of the curve for a short duration.

**Figure 15 micromachines-13-01606-f015:**
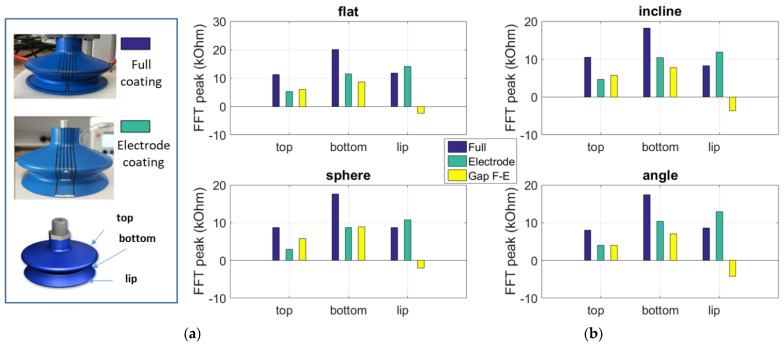
Influence of electrodes: (**a**) two SCs with full coating and electrode coating, (**b**) FFT peak value (fundamental harmonic) using four surface configurations.

**Figure 16 micromachines-13-01606-f016:**
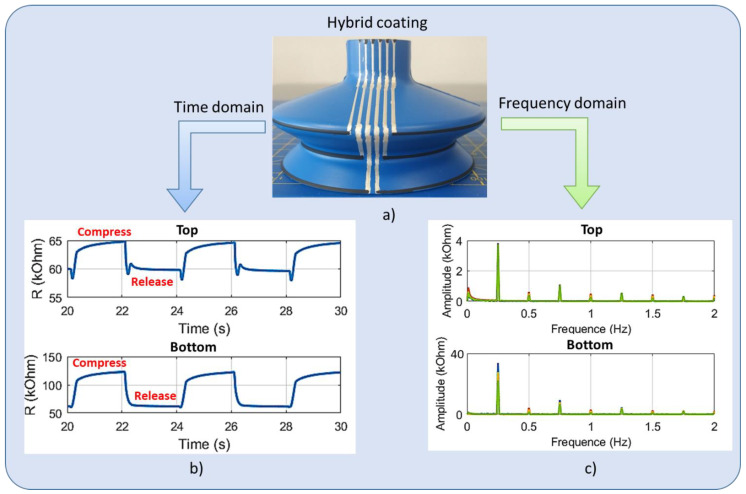
Time and frequency domain analyses of the top and bottom resistances for the hydride-coating suction cup: (**a**) printed sample, (**b**) resistance versus time, and (**c**) resistance change through FFT analysis.

**Figure 17 micromachines-13-01606-f017:**
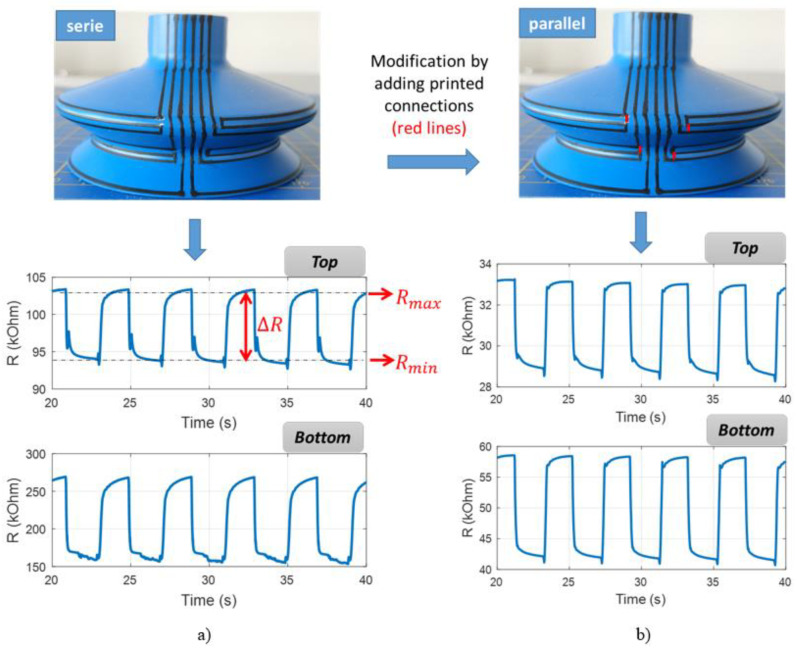
Time evolution of the three-circle resistances in (**a**) serial connection or (**b**) parallel connection coated on a 3D suction cup.

**Figure 18 micromachines-13-01606-f018:**
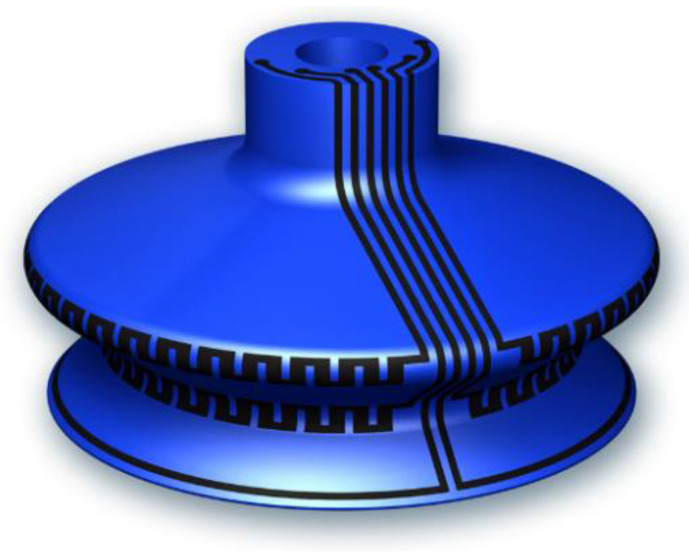
Design of SC with crenelated patterns.

**Figure 19 micromachines-13-01606-f019:**
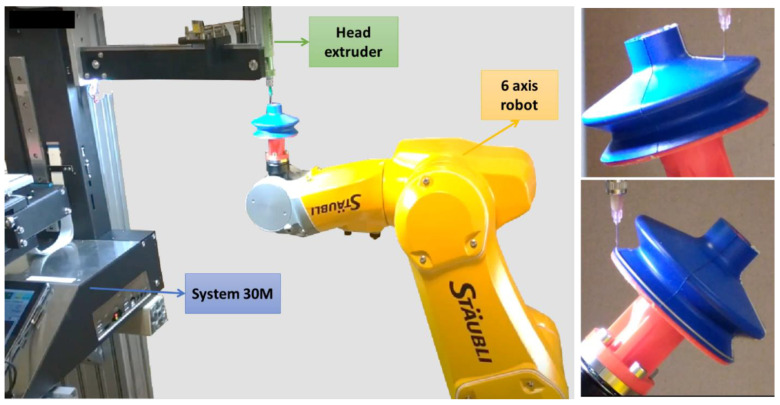
Advances in 3D printing process using a 6-DOF robotic arm.

**Table 1 micromachines-13-01606-t001:** Dimension and force characteristics of a Ø78 suction cup.

Lip diameter (ØA)	78 mm	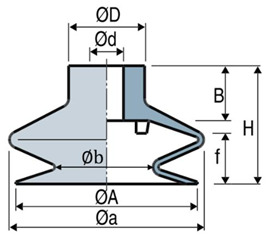
SCstroke (f)	14 mm	
Depth of housing fitting (B)	20 mm	
Height (H)	46.8 mm	
External neck diameter (ØD)	25 mm	
Internal neck diameter (Ød)	12 mm	
Internal volume (V_i_)	76 cm^3^	
Top bellow diameter (Øa)	83 mm	
Bottom bellow diameter (Øb)	43.7 mm	
Tensile force (F_t_)	110 N	
Slipping force (F_g_)	55 N	

**Table 2 micromachines-13-01606-t002:** Properties of the conductive inks.

	First Ink	Second Ink
Viscosity (cps)	30,000–35,000	57,000
Useful temperature range (°C)	−55 to 120	−50 to 100
Curing temperature range (°C)	50–180	50–120

**Table 3 micromachines-13-01606-t003:** Dimension and electrical measurements of the conductive inks.

	First Ink	Second Ink
Width *d* (mm)	1.18	0.9
Length *L* (mm)	60	60
Thickness *e* (µm)	36	22
Electrical resistance (kΩ)	10	34

**Table 4 micromachines-13-01606-t004:** Comparison of the absolute and relative resistance variation of three SC coating designs.

	Type of Suction Cup		
	1-Circle Hybrid Coating	3-Circle Serial Coating	3-Circle Parallel Coating
Top bellows	∆*R* = 7 kΩ	∆*R* = 10 kΩ	∆*R* = 4.5 kΩ
	∆_*m*_ = 5%	∆_*m*_ = 5.1%	∆_*m*_ = 7.3%
Bottom bellows	∆*R* = 60 kΩ	∆*R* = 110 kΩ	∆*R* = 16 kΩ
	∆_*m*_ = 35.2%	∆_*m*_ = 26.2%	∆_*m*_ = 16%

## Data Availability

Not applicable.

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
