# Peer review of "Extrusion-Based 3D Printing of Stretchable Electronic Coating for Condition Monitoring of Suction Cups"

_micromachines, 2022, doi:10.3390/mi13101606_

Round 1

Reviewer 1 Report

The authors have done excellent work on stretchable electronic coating for condition monitoring suction cups. 

I am very much satisfied with this research work. And I am suggesting some queries regarding this manuscript.

1. In the introduction section, the detail about the literature regarding Polymer 3D printing is insufficient.

2. what is the standard followed for tensile testing 

3. what is the strain rate given for tensile testing. 

4. Section 4.1 how did you differentiate the two inks by SEM/EDS analysis? Explain from a technical point of view. 

Reviewer 2 Report

This manuscript titled “Extrusion-based 3D printing of stretchable electronic coating for condition monitoring suction cups” by Cottinet et, al. demonstrated fabrication and characterization of printed sensors on suction cups. The results in this manuscript are well organized and sufficient evidences are provided to support the conclusion. However, the manuscript is not presented in a scientific way and the value is hard to evaluate because some information related to the suppliers is confidential. I personally do not think it is appropriate to be published in this journal and hereby recommend rejection of this manuscript.  Below are some of my suggestions to the authors.

1.      When you are mentioning “coating”, I think it is referring to “printing”? It can be confusing since coating usually refers to deposition of materials into a film covering certain area.

2.     For Section 2.3 the printing process, you do not need to introduce the capability of the 3D printer. It makes the manuscript feel like the advertisement of the 3D printer.

3.     You mentioned the FEM results were from your industrial partner. Please be sure you handle the authorship appropriately. If they provided the results, they should be considered to have significant contribution to the manuscript.

4.     How do you cure the ink? For the samples printed on 2D planar structure and 3D curved structure, are they showing different performances?

Reviewer 3 Report

In the submitted paper, a method based on the resistance measurement of the printed conductive coatings was employed to monitor the condition of a suction cup. I would like to recommend this manuscript be accepted for publication in present journal. My one question is that if there is initial mechanical strain within the SC, then the present monitoring method still works?

Round 2

Reviewer 2 Report

Authors have addressed my comments. I recommend acceptance of this manuscript.